# Paternal exposure to benzo(a)pyrene induces genome-wide mutations in mouse offspring

Marc A. Beal[1,2], Matthew J. Meier[2], Andrew Williams[2], Andrea Rowan-Carroll[2], Rémi Gagné[2], Sarah J. Lindsay[3], Tomas Fitzgerald[3], Matthew E. Hurles[3], Francesco Marchetti [2] & Carole L. Yauk[2]

Understanding the effects of environmental exposures on germline mutation rates has been a decades-long pursuit in genetics. We used next-generation sequencing and comparative genomic hybridization arrays to investigate genome-wide mutations in the offspring of male mice exposed to benzo(a)pyrene (BaP), a common environmental pollutant. We demonstrate that offspring developing from sperm exposed during the mitotic or post-mitotic phases of spermatogenesis have significantly more de novo single nucleotide variants (1.8-fold; $P < 0.01$) than controls. Both phases of spermatogenesis are susceptible to the induction of heritable mutations, although mutations arising from post-fertilization events are more common after post-mitotic exposure. In addition, the mutation spectra in sperm and offspring of BaP-exposed males are consistent. Finally, we report a significant increase in transmitted copy number duplications ($P = 0.001$) in BaP-exposed sires. Our study demonstrates that germ cell mutagen exposures induce genome-wide mutations in the offspring that may be associated with adverse health outcomes.

[1] Carleton University, Ottawa, Ontario K1S 5B6, Canada. [2] Environmental Health Science and Research Bureau, Healthy Environments and Consumer Safety Branch, Health Canada, Ottawa, Ontario K1A 0K9, Canada. [3] Wellcome Trust Sanger Institute, Wellcome Trust Genome Campus, Hinxton, Cambridge CB10 1SA, UK. Correspondence and requests for materials should be addressed to F.M. (email: francesco.marchetti@canada.ca) or to C.L.Y. (email: carole.yauk@canada.ca)

There is growing recognition of the importance of heritable mutations across a broad array of human genetic diseases. However, germline mutations are rare ($10^{-8}$ per nucleotide)[1], and studying these mutation events has been difficult historically because of technological limitations. Consequently, there are no accepted human germ cell mutagens, although many rodent germ cell mutagens have been identified[2].

Advances in genomic technologies, specifically next-generation sequencing (NGS) and comparative genomic hybridization arrays (aCGH), have allowed de novo mutation rates, and spectra to be studied at the genome-wide level[3,4]. These technologies have been applied to confirm previous observations that paternal age is a strong determinant of de novo mutation rate, with older fathers passing on more mutations to their children than their younger counterparts[5,6]. Hence, paternal age is now considered the first confirmed human germ cell mutagen. Similarly, more recent evidence suggests that maternal age[7] and dioxin[8] also increase the mutational burden in humans. These findings emphasize the need to identify both the endogenous and exogenous factors that may be contributing to human germ cell mutagenesis and genetic diseases.

Many paternal mutations are assumed to occur spontaneously during DNA replication in self-renewing spermatogonial stem cells. As a result, the contributions of paternal environmental exposures to mutation formation are likely underestimated. We have previously demonstrated that even small (e.g., 10%) increases in germline mutation rates can have serious public health implications[9]. Pedigree studies in mice have demonstrated that exposures to DNA-damaging agents lead to the transmission of induced de novo mutations[10]. New whole-genome approaches in animal models support the hypothesis that paternal environmental exposures contribute to heritable mutagenesis. For example, Adewoye et al.[11] demonstrated using aCGH and NGS that paternal radiation exposure induces large deletions and clustered mutations (multisite mutations) in the genomes of their offspring. This provided proof of principle that whole-genome approaches can be used to investigate the role of environmental exposures in inducing de novo genome-wide mutations through the germline.

Here, we applied genomic tools to study the consequences of paternal exposure to benzo(a)pyrene (BaP), a ubiquitous environmental pollutant, on the mouse germline. BaP is an IARC class 1 carcinogen[12] that is metabolically activated by cytochrome p450s to produce reactive metabolites, such as 7,8-dihydrodiol-9,10-epoxide (BPDE)[13], that form bulky DNA adducts. These DNA lesions can cause strand breaks, and replication of DNA unrepaired adducts leads primarily to G to T mutations[14]. We previously showed that the same exposure used in this study induces a fourfold increase in mutation frequency in a transgenic reporter gene[15], and a twofold increase in microsatellite mutations[16,17] in mouse sperm. Here, we directly compare the spectrum and frequencies of mutations measured in exposed germ cells from these previous studies to mutations found in the offspring and demonstrate that paternal exposure to BaP induce genome-wide mutations in the offspring. The similarity in the mutation spectra observed in the sperm of exposed males and their offspring supports the heritability of environmentally induced germ cell mutations.

## Results

**Experimental approach.** We investigated the effects of paternal BaP exposure on genome-wide heritable mutation frequencies and spectra in the unexposed offspring (experimental design shown in Fig. 1). MutaMouse males received 0 or 100 mg/kg of BaP dissolved in olive oil via oral gavage for 28 days, and were mated with C57Bl/6J females to facilitate the assessment of the parental origin of mutations. Matings occurred 3 and 42 days after the last daily exposure, and no reduction in litter size was observed in matings with BaP-exposed males versus control matings (data not shown).

The two mating points were selected to investigate the induction of mutations during phases of spermatogenesis that differ in DNA repair activity. The 3-day time point was chosen to

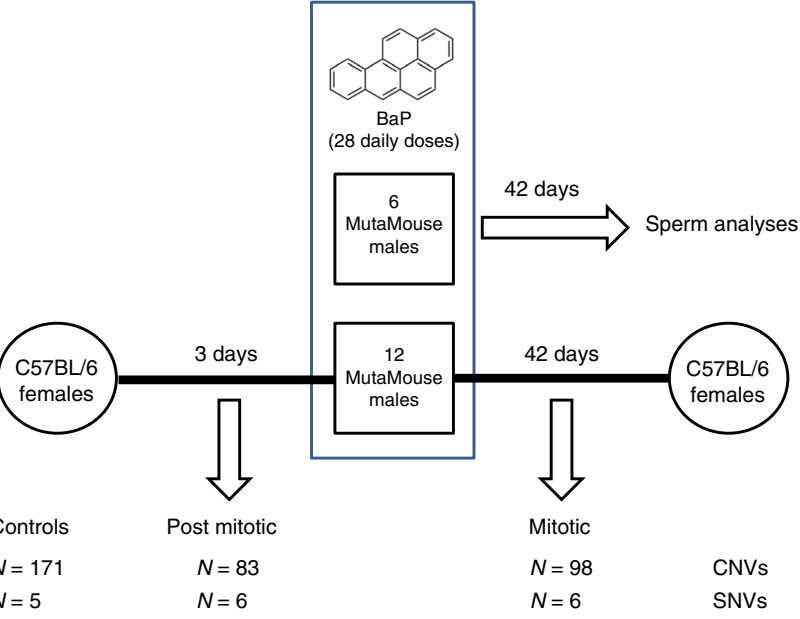

**Fig. 1** Experimental design. Twenty MutaMouse males received 28 daily doses of 0 or 100 mg/kg BaP. The sperm from six controls and six BaP-exposed males were collected 42 days after exposure and analyzed for the presence of *lacZ* mutations[15] and microsatellite mutations[17], and for mutation spectrum analysis[27]. Twelve randomly selected control and BaP-exposed males were mated with untreated C57Bl/6 females 3 and 42 days after the end of the exposure and their offspring analyzed for the presence of single nucleotide variants (SNVs) and copy-number variants (CNVs)

study offspring developing from sperm exposed to BaP during the post-mitotic phase of spermatogenesis (spermatocytes and spermatids), which lacks DNA synthesis and has reduced repair capabilities[4]. These processes are required to fix BaP lesions into mutations; consequently, no mutation induction should occur in these cells[18]. Indeed, we showed in a parallel study that the exposure used herein causes a fourfold increase in mutant frequency in the *lacZ* transgene of sperm exposed as spermatogonia, but not as spermatids[15]. However, sperm DNA lesions can be converted into mutation in the egg following fertilization[19,20], when the oocyte's DNA repair activity[21] is unable to repair all lesions. Replication of unrepaired BaP lesions in sperm DNA during the first cellular divisions would convert those lesions into mutations in the developing embryo that would appear as embryonic mutations with a variant allele fraction (VAF) of ~0.25 (see the Methods section; Supplementary Fig. 1). The latter time point was chosen to investigate offspring developing from sperm exposed during the mitotic phase (stem cells and differentiating spermatogonia) of spermatogenesis, which has active DNA repair[22], when mutations can be fixed in the developing germ cells and are already present before the sperm fertilizes the egg. These mutations are expected to appear as de novo mutations with a VAF of ~0.5. Thus, we hypothesized that BaP exposure would lead to increases in the number of induced mutations in the offspring at both mating times, but with a different proportion of de novo versus embryonic mutations.

Two approaches were used to quantify and characterize genome-wide mutations occurring before and after fertilization. The first approach involved whole-genome sequencing (WGS) of family quintets (sire, dam, and three offspring) using spleen DNA to identify all detectable mutation events. The whole genomes of two control families 42 days after the last oral gavage, and two treated families from both mating periods, were sequenced. Thus, a sample size of six offsprings per treatment group was studied, except for one control family where sequencing quality metrics were below thresholds for one offspring sample and the data were not used. The two BaP-treated sires used for mating in the post-mitotic group were also used for mating in the mitotic group. Putative mutations were cross-validated using Illumina® targeted re-sequencing. For the second approach, CGH arrays were used to identify de novo copy-number variations (CNVs), again by comparing offspring to parental DNA. In total, liver DNA from 171, 83, and 98 offsprings were analyzed for de novo CNVs in control, post-mitotic, and mitotic groups, respectively. All CNVs were validated using real-time PCR and by mate-pair NGS.

Broadly, the WGS and CNV results of this study were used to (1) determine whether paternal BaP exposure increases the number of inherited mutations; (2) explore whether pre-mutational lesions in sperm are converted into mutations in the offspring after fertilization; and (3) contrast mutagenic effects in germ cells with those occurring in offspring.

**De novo single-nucleotide variants and indels**. We used WGS of two families from the control, and post-mitotic and mitotic exposure groups to investigate the induction of single-nucleotide variants (SNVs) and indels following paternal exposure to BaP. WGS followed by targeted re-sequencing identified 275 true mutations (average 18 per BaP animal; 11 per control; Supplementary Data 1). In total, 190 validated variants were de novo (average 13 per BaP animal; seven per control; Supplementary Data 1), 73 were early embryonic (average five per BaP animal, three per control; Supplementary Data 2), six were germline mosaics in the parents (three mutations shared by two siblings), and six were somatic mosaics in the parents. The spontaneous de novo mutation frequency in this study was $3.1 \pm 0.7 \times 10^{-9}$,

which is lower than other estimates using different technologies[11,23], but is within range. The proportion of mosaic/embryonic mutations among SNVs/indels was ~25%, which is highly consistent with previous measurements in mice with a larger sample size (non-peer reviewed study)[23]. Four de novo and three embryonic mutations were identified as nonsilent mutations within genes by Ensembl Variant Effect Predictor[24] (four treated; three control; Supplementary Data 3). Haplotype phasing of the mutations, where possible, indicated that most mutations were paternal in origin (22 paternal, five maternal; Supplementary Data 4).

Overall, the number of de novo mutations passed onto offspring was higher in the BaP group (65 mutations in six offspring from the post-mitotic exposure group, 88 mutations in six offspring from the mitotic exposure group) compared with olive oil-treated controls (37 mutations in five animals). The total number of de novo mutations in the BaP group was 1.7-fold higher than controls (adjusted $P = 0.036$ using generalized estimating equations (GEE) for individual offspring effects). Furthermore, BaP exposure elevated the total number of mutations (i.e., both embryonic and de novo) per offspring by ~1.8-fold (adjusted $P = 0.04$; GEE; Table 1; Fig. 2; 103 + 110 BaP mutations in 12 animals vs. 50 control mutations in five animals). When analyzed based on the window of spermatogenic exposure, we observed that de novo mutations were significantly increased only after exposure of the mitotic phase (adjusted $P = 0.03$, GEE), while embryonic mutations were significantly increased only after post-mitotic exposure (adjusted $P = 0.038$, GEE). Taken together, these results support the expected mechanism of mutation induction and provide strong evidence that BaP-induced sperm mutations are heritable and genome-wide.

A large portion (10/12) of offspring in the BaP-exposed group had a higher number of mutations than the mean number of mutations in controls. Moreover, there was a large degree of variation among the offspring of BaP-treated males, with seven of the mice having numbers of mutations outside the range of control mutations and five mice being within the control range. This observed variability in mutations occurred within litters, indicating that this was not due to differences in exposure levels among sires but to different amount of damage among the fertilizing sperm. We estimate, based on previous measurements of BaP adducts[25], that the dose and frequency of the exposure would lead to ~1000 bulky DNA adducts per genome. The detection of the de novo SNVs indicates that some of these adducts elude DNA repair, while the observation that only some animals have elevated mutation levels suggests that BaP is not evenly distributed across the germ cell population during exposure. Alternatively, different cells may respond differently to exposures, possibly through adaptive upregulation of different DNA repair genes[26]. Hence, only a certain percent of animals would have BaP-associated SNVs and indels or show an overall increase in genome-wide mutation count.

The mutation spectra of the seven offspring with the highest numbers of mutations (Fig. 2) revealed that these extra mutations were consistent with the mutation spectrum of BaP (i.e., increased frequency of G:C → T:A, G:C → C:G) observed in the sperm and bone marrow of exposed males[14,27]. Furthermore, there was also an increase in the number of G:C → A:T mutations (Fig. 2). Overall, the mutation spectrum in the affected animals in the exposed group was significantly different than that observed in unaffected animals and controls (Fisher's exact $P = 0.0009$).

Next, we explored how our BaP-induced mutation spectrum aligns with mutation signatures found in human cancers from the "Catalogue Of Somatic Mutations In Cancer" (COSMIC) database[28]. About 23% of SNVs (total de novo and embryonic SNVs across animals; Supplementary Data 5) in the seven

**Table 1 De novo mutations in the offspring of the control and BaP-treated animals**

| Mutation type | Exposure group | Sample size[a] | Number of mutations | Mutations per offspring (95% CI) | Ratio to control[b] | P-value[d] | Holm–Sidak adjustment[d] |
|---|---|---|---|---|---|---|---|
| De novo SNVs/indels | Control | 5 | 37 | 7.4000 (5.4845–10.7368) | — | — | |
| | Post-mitotic | 6 | 65 | 10.8333 (8.8010–14.5347) | 1.46 | 0.074 | 0.074 |
| | Mitotic | 6 | 88 | 14.6667 (12.3822–19.0208) | 1.98 | **0.010** | **0.030** |
| | All BaP | 12 | 153 | 12.7500 (11.3787–15.7242) | 1.72 | **0.018** | **0.036** |
| Embryonic SNVs/indels | Control | 5 | 13 | 2.6000 (1.4573–4.6801) | — | — | |
| | Post-mitotic | 6 | 38 | 6.3333 (4.7177–9.1505) | 2.44 | **0.013** | **0.038** |
| | Mitotic | 6 | 22 | 3.6667 (2.4188–5.8436) | 1.41 | 0.220 | 0.220 |
| | All BaP | 12 | 60 | 5.0000 (4.0163–6.7747) | 1.92 | **0.050** | **0.097** |
| Total SNVs/indels | Control | 5 | 50 | 10.0000 (7.8128–13.8776) | — | — | |
| | Post-mitotic | 6 | 103 | 17.1667 (14.7495–21.9153) | 1.72 | **0.001** | **0.003** |
| | Mitotic | 6 | 110 | 18.3333 (15.8608–23.2596) | 1.82 | **0.005** | **0.005** |
| | All BaP | 12 | 213 | 17.7500 (16.2591–21.3690) | 1.78 | **0.002** | **0.004** |
| Large deletions | Control | 171 | 3 | 0.0175 (0.0036–0.0527) | — | — | |
| | Post-mitotic | 83 | 1 | 0.0120 (0.0001–0.0716) | 0.69 | 0.604 | 0.604 |
| | Mitotic | 98 | 0 | 0.0000 (0.0000–0.0323) | 0.00 | 0.255 | 0.587 |
| | All BaP | 181 | 1 | 0.0055 (0.0001–0.0338) | 0.31 | 0.290 | 0.496 |
| Duplications/insertions | Control | 171 | 0 | 0.0000 (0.0000–0.0188) | — | — | |
| | Post-mitotic | 83 | 3 | 0.0361 (0.0080–0.1053) | 5.60 | **0.034** | 0.099 |
| | Mitotic | 98 | 2 | 0.0204 (0.0011–0.0759) | 4.04 | 0.132 | 0.132 |
| | All BaP | 181 | 5[c] | 0.0276 (0.0101–0.0648) | 3.45 | **0.035** | 0.069 |

[a]Sample size refers to the number of offspring analyzed in each group
[b]In the cases where the ratio was undefined (control rate = 0) the upper limits of the 95% CI were used to estimate the ratio
[c]Two of the CNVs occurred as a single event in one of the mice in the post-mitotic group. Thus, we report the number of mutations in this group as 3 and not 4, and the total count is reported conservatively as 5 instead of 6
[d]Statistically significant P-values are indicated in bold

offspring with elevated numbers of mutations were aligned with COSMIC Signature 4. This signature is derived from mutations in human lung cancers of tobacco smokers[29], and is the major signature found after exposure to BaP[30]. There was no association between Signature 4 and mutations in the mice with lower numbers of mutations, supporting that the extra mutations in the affected animals resulted from BaP exposure. Overall, these findings provide strong support that the additional mutation burden is due to the presence of the bulky BPDE adducts at guanine nucleotides, confirming the effects of BaP on both de novo and embryonic mutation induction.

Four of the seven animals with elevated mutation levels developed from sperm exposed to BaP during the post-mitotic phases of spermatogenesis (Fig. 2). This provides further support to the hypothesis that DNA damage occurring in later phases of spermatogenesis can lead to mutation formation in the developing embryo. Interestingly, the animal with the most embryonic mutations (32CM1) had 12 BaP-type mutations in total with evidence of some clustered mutations. Specifically, this mouse had three G:C → T:A mutations (VAF: 0.18–0.21) within a 45 -kb window on chromosome 12 and two G:C → T:A mutations (VAF: 0.31–0.35) within a 438 kb window on chromosome 17. Considering that the average spacing of mutations across all animals was 8.9 Mb, it is unlikely that these mutations occurred close together by chance alone[31]. Furthermore, the mutational fingerprint of these events indicates that they were likely a result of BaP adducts, and the similarities in VAFs indicate that the mutations formed concurrently as that genomic region was replicated during early embryonic development. Previous observations suggest that the earliest cell divisions during embryogenesis have the highest spontaneous mutation rates compared with any other germline stage[32] and confirmed in non-peer reviewed work[33]. The evidence presented here also suggests that these early stages represent a sensitive window by which DNA damage leads to genetic mosaicism in the developing organism.

The overall increase in mutation count in the BaP-exposed offspring was unexpectedly lower than the increase in mutation

frequency observed in sperm following the exact same exposure (approximately twofold vs. fourfold, respectively)[15]. There are a few possibilities that could explain this: (1) the small number of meioses analyzed in the pedigree study versus the sperm work (four meioses per group for the pedigree study, >1,00,000 per group in the sperm study) may lead to lower accuracy in the pedigree approach; (2) negative selection against germ cells and embryos with the highest amounts of DNA damage decreased the number of detectable mutations in BaP offspring; (3) mutation induction in the transgene analyzed in sperm does not provide an accurate representation of the mutation induction in the rest of the genome, which could occur if there are sequence or region biases that make this transgene a mutational hotspot; and (4) a disproportionate amount of BaP mutations were missed compared with the control using the WGS approach. The latter would most likely be due to the high probability of BPDE adducts forming in regions rich in CG nucleotides that are difficult to sequence.

A combination of technical issues with sequencing and stringent bioinformatics filtering cannot be ruled out to explain the lower than expected number of mutations retrieved. For example, indels accounted for 33% of the BaP mutations detected in sperm[27], and only a few indels were detected in the genomes of the offspring. Based on the functional mutations observed in the lacZ gene in sperm of BaP-exposed mice[27], we estimated the total number of expected mutations in the genomes of offspring from the control and BaP-treated sires. In total, there were ~60 mutations detected in 11.5 Gb and 300 mutations in 10.9 Gb for the control and BaP sperm, respectively. Assuming a random distribution of mutations, there should be ~32 and 170 mutations per diploid genome in the control and BaP mice, respectively. By this logic, 31% of possible control mutations were detected by WGS, but only 10% were detected in the BaP group. Thus, although we have shown that BaP induces genome-wide mutations, the discrepancy between the expected and observed number of mutations suggests that the full extent of the damage induced by BaP remains uncertain.

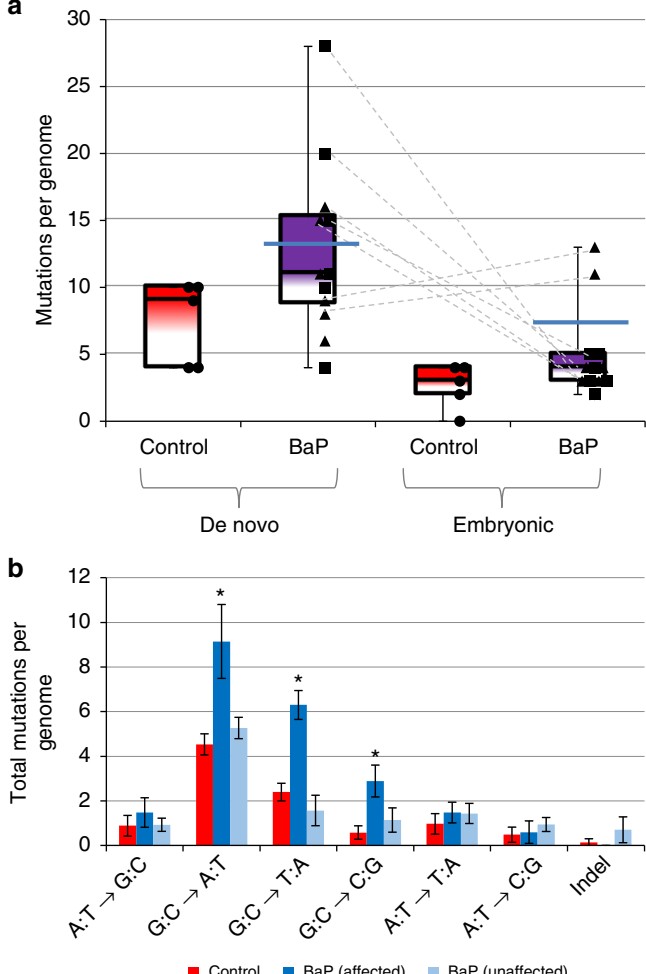

**Fig. 2** Mutation counts and spectra in the control and BaP offspring as measured by WGS. **a** Each circle, triangle, and square represent one offspring from the control (*n* = 5 animals), post-mitotic (*n* = 6 animals), and mitotic (*n* = 6 animals) exposure groups, respectively. The dashed lines connect the embryonic and de novo mutations for the respective animals. For analytical purposes, we called the offspring above the dark blue lines (above the means of the BaP-exposed group) as "affected" by BaP, and those below "unaffected." **b** Separating the BaP offspring into two groups (affected, and unaffected) shows that the additional mutations in the affected mice match the mutation fingerprint of BaP indicated with asterisks (G:C → T:A, G:C → C:G; Supplementary Data 1; Supplementary Data 2; generalized estimating equations $P \leq 0.0027$). Furthermore, there is an increase in G:C → A:T mutations (shown in asterisks, generalized estimating equations $P = 0.0010$). In our previous analyses, we observed a decrease in proportion of this mutation type, but the absolute mutation frequency is increased[14,27]. Overall, the mutations induced by BaP occur at guanine nucleotides as expected. Source data can be found in Supplementary Data 8

There were limitations that prevented the estimation of mutation rate for each sire. First, the work was somewhat confounded as multiple offspring per male were sampled. Second, there was a large degree of variability among siblings with no apparent litter effects. However, the variation in the number of BaP-induced mutations in each offspring appears to be a biological and not a technical result. Future work that precisely measures mutation frequency in individual sperm using techniques such as error-corrected sequencing[34] could shed light on this

type of variability and its influence on chemically induced de novo mutation rates.

**de novo CNVs**. We applied stringent filters to identify CNVs from the aCGH analysis of pedigrees and then validate each candidate with real-time PCR and breakpoint sequencing (Fig. 3; see the Methods section). This resulted in a final set of ten confirmed de novo and two germline mosaic CNVs (Supplementary Data 6). The spontaneous de novo deletion CNV frequency observed in our control mice (1.8%; 3/171) was comparable with previous estimates (1.5%; 2/136) using combined data from two published mouse studies[11,35]. The frequency of deletion CNVs in the treated group was 0.5–1% (1/83, 0/98 for the two time points, respectively; 1/181 total); hence, there was no evidence for induction of deletions (Table 1). In contrast, there was an increase in duplication CNVs in the treated group (4/83, 2/98 for the two time points, respectively; 6/181 total) relative to the control animals (0/171; Fig. 4; Table 1; Supplementary Data 6). Two of the duplication CNVs occurred in the same animal, thus, the total number of CNVs was conservatively counted as five. Using this number, the increase in duplications in the BaP group was at borderline of significance (Fisher's exact test, adjusted $P = 0.08$). However, the five BaP-induced duplication events occurred in five different litters from four sires (males 32, 33, 37, and 39). Male 32 had an offspring with a duplication CNV at both time points, one involving chromosome 3 and the other chromosome 16; while male 33 had a male offspring with a duplication involving chromosome 5 and a female offspring with a deletion involving chromosome 3 within the same litter (Supplementary Data 6). Using sires as the biological unit, and considering the number of offspring analyzed for each sire (Supplementary Data 7), yields a statistically significant increase in the frequency of duplication CNVs (0.0278 vs. 0; Fisher's exact test, adjusted $P < 0.001$) demonstrating that this result is not due to clustering within a litter or within a BaP-exposed male. Indeed, the spontaneous frequency of duplications appears to be rare compared with deletions, as only a single duplication was detected from 93 control animals from Adewoye et al.[11], 43 control animals from Arlt et al.[35], and 171 animals in this study. Thus, although we have recovered few duplications overall, this increase in the exposed mice is unlikely to be due to chance alone.

Duplication events have potential phenotypic consequences as they can disrupt gene dosage and homeostasis. Furthermore, a duplicated sequence can be inserted within a gene and disrupt that gene's function. Therefore, we used sequencing to characterize the breakpoints of the CNVs to elucidate both the nature of these BaP-induced duplications (tandem vs. inserted) and the mechanism for their formation. We found that three of the six duplication CNVs were insertions at other locations in the genome (different chromosomes). Two of these insertion CNVs were present in the same animal (Supplementary Fig. 2A), but on separate chromosomes (4 and 10; Table 1). Sequencing revealed that the two CNVs were joined together as an insertion into a long interspersed nuclear element (LINE: L1Md_T). The exact location of this insertion could not be determined due to the repetitive structure and abundancy of this LINE. The other insertion (Supplementary Fig. 2B) involved a region from chromosome 16 being copied into a satellite sequence (GSAT_MM) on another chromosome (most likely chromosome 2 based on the 97% sequence identity at the breakpoint). Remarkably, both breakpoints had nucleotide similarity to the satellite sequence indicating that normal replication of the satellite resumed after insertion of the chromosome 16 donor sequence. The other three duplication CNVs occurred in tandem

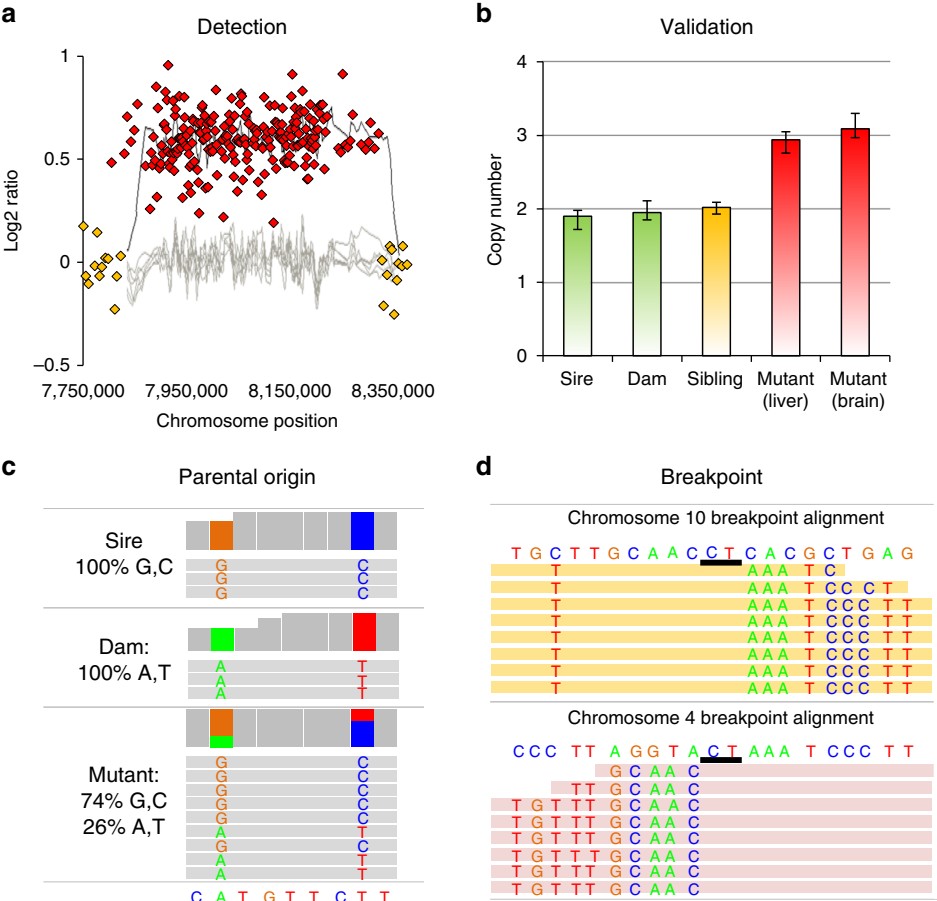

**Fig. 3** Identification and characterization of de novo CNVs. **a** Example of a 533-kb duplication. Red diamonds show the DNA probes with higher offspring signal (elevated Cy5 at CNV locus) and the yellow diamonds show the DNA probes where offspring DNA is the same quantity as the parental DNA. The red glowing line shows the average $\log_2$ ratio (0.61) for the mouse with a CNV while the yellow glowing lines show the average $\log_2$ ratio for the six litter mates. The $\log_2$ ratio of zero indicates no change in copy number. **b** The copy number measured by qPCR revealed that the putative CNV was a true de novo event because it was present in multiple tissues with a copy number of three, and the CNV was not present in the parents or sibling. **c** Mate-pair sequencing revealed that the CNV came from the sire because the paternal SNPs were present at a higher allele fraction within the CNV locus. **d** Mate-pair sequencing identified the breakpoint locations by mapping discordant mate-pairs against the genome (mapping at chromosome 10 in orange and 4 in pink). Split reads mapping to both chromosome locations simultaneously allowed for the breakpoint sequence to be determined. In this example, the CNV was facilitated by microhomology of two base pairs (CT)

(Supplementary Fig. 2C, D). One of the tandem duplications was a complex event involving a 28 -bp insertion from a region 16 kb upstream from the nearest breakpoint (Supplementary Fig. 2C). The orientation of the discordant mate pairs also signified that this event was associated with an inversion, further highlighting the complexity of this mutation and the region where it occurred. Given that some CNV breaks in our study were within satellite DNA, and that only large CNVs can be detected in such low-complexity sites as a result of the aCGH design, it is probable that CNVs of moderate size in repetitive regions of the genome were undetected.

All of the de novo CNVs with detectable breakpoints appear to have been mediated by overlaps consisting of short microhomologies of 1–6 bp between sequence breakpoints (Supplementary Data 6). These events are similar to the CNVs detected in irradiated human fibroblasts[36] and in the germline of mice exposed to hydroxyurea[35] where 1–8 -bp homologies were observed at most breakpoint junctions. The CNVs with microhomology preclude involvement of homologous recombination repair (HRR) because the sequence homology required for HRR is >200 bp[37]. The model that best explains these mutation events is microhomology-mediated break-induced replication

(MMBIR)[38,39], which is an alternative DNA repair pathway invoked to repair double-strand DNA breaks.

The results of this study and previous observations[40] suggest that deletions occur more commonly in the mouse germline, and that duplication CNVs form infrequently in the absence of replication fork collapse near DNA damage. Indeed, it has been postulated that MMBIR favors the formation of gains over losses, and that an enrichment of duplications represents a mutational signature that may require environmental stress[41]. In this experiment, BPDE was likely the main environmental stress leading to replication fork collapse and subsequent MMBIR. In addition, BaP metabolism creates reactive oxygen species[42] capable of inducing replication fork collapse via base excision repair mechanisms. An excess of collapsed replication forks in the presence of BaP damage may lead to Rad51 depletion, inhibiting strand invasion into the homologous sequence required for HHR[38], leading to pre-fertilization CNVs in spermatogonia. Likewise, mature sperm lack homologous chromosomes to guide repair, leading to post-fertilization CNVs in the developing embryo. In lieu of a homologous template, the collapsed fork invades an active replication fork within physical proximity using the microhomology as a primer. This hypothesis is supported by

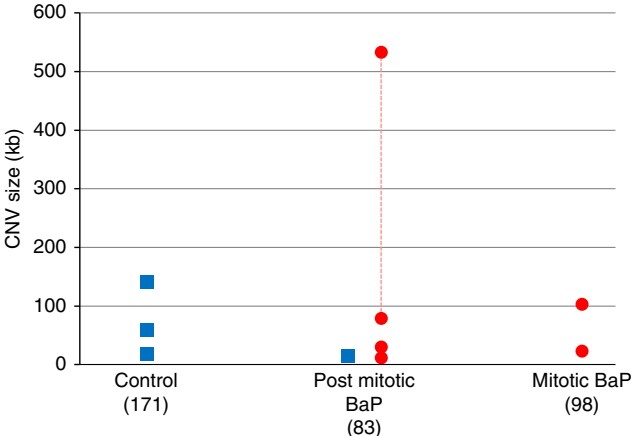

**Fig. 4** Size and types of de novo CNVs characterized in control and BaP animals. Blue squares indicate deletions and red circles indicate duplications or insertions. The red dotted line signifies that the two insertion CNVs are from the same animal and were derived from the same mutation event. Post-mitotic and mitotic refer to matings occurring 3 and 42 days after the end of BaP exposure, respectively. Source data can be found in Supplementary Data 9

previous observations using a bacterial model investigating BPDE adducts[43]. Specifically, DNA segments containing BPDE adducts were more likely to be deleted or replaced with an alternative sequence (>42 bp) than segments without adducts (no alterations observed). Similar to the CNVs detected here, the breakpoints of the alterations showed microhomology (most often 2 bp) and the alterations occurred independent of RecA (Rad51 homolog). Taken together, these findings suggest that BPDE adducts lead to fork collapse and instability, favouring MMBIR, which copies DNA from another source at the site of fork collapse.

There are limitations to our de novo CNV analysis. First, the ability to detect CNVs that were present as mosaics was limited (see Supplementary Discussion, Supplementary Figs 5, 6). Second, assigning the parental origin of each CNV was challenging. Parental origin could only be assigned for four of the nine de novo CNVs. All four were paternal in origin. Thus, total (rather than paternal exclusive) CNV frequencies were compared between the control and BaP animals due to the inability to determine the parental origin for all CNVs. Because CNV occurrence per sire is rare, it is difficult to precisely estimate the rate of CNV formation to compare rates across fathers. In order to determine CNV frequencies for individual sires, single-sperm analyses should be applied. Further work is required to better understand the relationship between DNA adducts and CNV duplication/insertion frequencies.

## Conclusions

We found that de novo mutations are elevated in the descendants of BaP-exposed males, consistent with increases in sperm mutation frequencies. De novo mutations can occur during exposure of stem cells and spermatogonia, and are transmitted to offspring. Our approach also enabled the detection of mutations originating as DNA damage in mature sperm that are undetected by existing sperm assays[27]. These results confirm that mature sperm represent a sensitive window of spermatogenesis by which exposure to DNA-damaging agents can lead to mutations in the embryo. An important finding from this study was that many of the heritable mutations are mosaics. Thus, caution is required in interpreting exposure-related mutations to ensure that the damage caused by the exposure is not underestimated through application of

filtering approaches that eliminate such mutation events. Overall, BaP exposure caused SNVs as well as non-recurrent duplication CNVs, the latter representing an endpoint that is not currently considered in genetic toxicology testing. This research demonstrates the promise offered by applying whole-genome approaches to the study of heritable mutagenesis. Our results have potential human health implication because BaP is a ubiquitous environmental pollutant and continuous exposure to low doses is unavoidable. Future work should explore the dose–response relationship to enable a more informed understanding of risk of various environmental, lifestyle, and dietary exposure to BaP. Our results also emphasize that men need to be aware long before family planning as well as in the weeks leading in to conception that their lifestyle is important for the future health of their children.

## Methods

**Benzo(a)pyrene exposure and mouse breeding**. All animal procedures were conducted according to conditions approved by the Health Canada Ottawa Animal Care Committee. The BaP exposure was conducted in parallel with the exposure done in previous studies[15,16,27]. Twenty MutaMouse males (BALB/C × DBA/2)[44], ~15 weeks of age, were exposed daily to BaP (Sigma-Aldrich, Oakville, Ontario, Canada) dissolved in olive oil (Sigma-Aldrich; highly refined, low acidity) via oral gavage (~150 µL) for 28 days (100 mg/kg body weight/day). This dose was selected based on the results of a pilot study that indicated that a higher dose would have resulted in significant toxicity and morbidity to the animals[15]. Twenty concurrent males received olive oil and served as negative controls. Three days after the last treatment, 12 randomly chosen control and BaP-treated males were mated with up to four C57Bl/6J females each over a 2-week period to produce offspring derived from germ cells exposed during the post-mitotic phase of spermatogenesis. This exposure window covered most of meiosis and the entirety of spermiogenesis (post-mitosis). Breeding was repeated 42 days after the last treatment with four new C57Bl/6J females per male to produce offspring from germ cells exposed during the mitotic phase of spermatogenesis. One of the controls and three of the BaP-treated males did not produce litters at both time points, while another BaP-treated male needed to be euthanized before the completion of the experiments. Thus, 11 controls and eight BaP-treated males produced litters that were used in this study.

The offspring produced from each mating were weaned at ~3 weeks. Each offspring was identified by a code that included a number for the father, a letter for the mother followed by either an M for male or F for female, and a second number indicating the sequence of weaning. For example, 3HF1 identifies the first female from the litter produced by mating male 3 with female H (note that the female letter code was the same for each male, thus, each male was mated with females identified with letters A to H). Three weeks after birth, litters and dams were anaesthetized with isoflourane and euthanized via cervical dislocation. Sires were euthanized after tissue samples from the offspring were collected. From each animal in the quintet, the liver, spleen, and brain were collected and stored at −80 °C. The six additional controls and BaP-treated males were euthanized at 42 days, and sperm analyzed for the presence of lacZ mutations[15], microsatellite mutations[17], and determining the mutation spectrum induced by BaP[27].

**Whole-genome sequencing**. Two of the families from each group used for array analysis were chosen at random for WGS. Each family consisted of a sire, dam, and three offspring. DNA was extracted from the spleen using the DNeasy Blood & Tissue Kit (Qiagen), and NGS libraries were built using NEBNext® Fast DNA Library Prep Set for Ion Torrent™ (New England Biolabs, Whitby, Ontario, Canada) according to the manufacturer's instructions. NGS was performed in-house using the Ion Chef™ and Ion Proton™ instruments (Thermo Fisher Scientific, Burlington, Ontario, Canada). Each genome was sequenced to an average autosomal coverage of ~31×.

**De novo mutation calling and filtering**. Sequence data were aligned to the GRCm38/mm10 reference genome using the Torrent Mapper, and the Torrent Variant Caller was used to call variants (Torrent Suite version 5.0.2). Genomic variant call files (gVCFs) were merged, and Mendelian violations (MVs) between offspring and parents were called and filtered to eliminate false-positive calls. Specifically, the loci ±10 bp of each putative MV were examined for allelic depth using samtools (v. 1.3). MVs were kept if they met the following criteria: the VAF for the variant in the parents was <5%, VAF in the offspring was >25%, the fraction of reads with mapping quality of zero did not exceed 0.1, read depths for all animals was at least seven, and the Fisher's exact test for strand bias had a P-value > 0.05. Variants occurring within repeat sequences and other related elements were removed from further analysis. A total of 2728 variant calls passed filtering (average of 160 per animal) and were selected for validation (Supplementary Data 1).

We considered variant allele fractions (VAFs) for each mutation to determine whether it originated as a de novo (already present in sperm) or embryonic mutation arising in early embryonic division. The timing and kinetics of the mutation occurrence during spermatogenesis will dictate if mutation fixation will occur before or after fertilization. For example, if a BaP adduct is formed and fixation leads to a mutation during mitosis in the parental gamete, the mutation will have a VAF of 0.5 in the developing embryo and will be called as de novo. If an adduct is formed after mitosis, when there is no DNA replication and repair is declining, the resulting mutation will most likely arise in the first embryonic division and will appear as embryonic with a VAF of 0.25. However, segregation and distribution of the mosaic mutation could lead to varying VAFs in the animal and its adult tissues. Previous studies have shown that three cells from the 64-cell stage go on to form the entire animal[45]. In these three cells, there would be six haploid genomes total (three paternal, three maternal). Therefore, the VAFs for embryonic events could potentially be 0.17 (1/6), 0.33 (2/6), or 0.5 (3/6) when there are one, two, or three mutant cells that form the animal, respectively (Supplementary Fig. 1).

**Cross-platform targeted re-sequencing validation.** Mutations were validated by Agilent Sure Select Target Enrichment as described (non-peer reviewed study)[23]. Bait design was successful for 2261 of the 2728 putative variants (Supplementary Data 1). After enrichment, all variants were multiplexed and sequenced to ~ > 300x coverage using Illumina sequencing. Duplicates were removed and in-house scripts were used to quantify read counts at each candidate site to determine if the variant was a true de novo, inherited, or a false positive based on parental and child VAFs (Supplementary Fig. 3). The Integrative Genomics Viewer[46] was used to manually inspect all confirmed mutations. Further filtering was applied to eliminate mosaic mutations (mutations called in siblings) and mutations that occurred during embryogenesis (VAF in offspring <39%; Supplementary Fig. 4).

**Mutation frequency estimation.** Bedtools[47] was used to estimate the percent of autosomal nucleotides in the diploid mouse genome (4.7 GB)[48] covered by NGS reads (≥10x coverage). Approximately 93% of the genome (4.4 GB) was covered by NGS reads in each animal. Repetitive DNA sequence regions, according to Repeatmasker[49], were removed (45% of diploid genome) leaving 2.4 GB of the autosomal diploid genome where mutations could be detected. True de novo mutations with a VAF > 39% in the autosomes were divided by the number of autosomal nucleotides to estimate mutation frequency.

**Comparative genomic hybridization and CNV detection.** Agilent SurePrint G3 Mouse CGH arrays with 1 M probes (G4838A; Agilent Technologies, Mississauga, Ontario, Canada) were used to detect de novo CNVs in the litters produced from mating MutaMouse sires with C57Bl/6J dams. The manufacturer's instructions were followed for each aCGH experiment (Agilent Oligonucleotide Array-Based CGH for Genomic DNA Analysis; version 7.3). Specifically, high-molecular weight DNA was extracted from the liver of each animal using the DNeasy Blood & Tissue Kit (Qiagen, Montreal, Quebec, Canada). Two of the BaP-treated males exhibited high degrees of somatic mosaicism in their liver, apparent by the presence of several false-positive CNV calls in the offspring, and therefore brain DNA was used instead. One microgram of offspring DNA was labeled with Cy5 dye (5190–3400 SureTag DNA Labeling Kit). Five hundred nanograms of DNA from each respective parent were mixed together and labeled with Cy3 dye. As positive controls, and to identify existing strain differences, two additional arrays were used to compare MutaMouse with C57BL/6J DNA. DNA labeled with Cy3 and Cy5 were mixed together for hybridization onto the CGH array.

The hybridized slides were scanned using an Agilent C scanner, and Feature Extraction v10.5 was used to extract the data. Following Feature Extraction, signal intensities (Cy3 and Cy5) were background subtracted and then normalized (Lowess normalization using Rank invariant probes). Nonuniform features (outlier probes with signals influenced by array position) were removed from analysis. The algorithms previously used by Adewoye et al.[11] were applied to detect autosomal CNVs as genomic segments with parent vs. offspring fluorescent signal log$_2$ ratios above or below threshold values as described previously[50]. Specifically, de novo CNV duplications/deletions were called if they had an absolute |log$_2$ ratio| > 0.4 (1.32-fold higher signal) and the CNV segment was a minimum of five consecutive altered probes. Probes had an average spacing of ~3,000 bp; therefore, this approach detected CNVs that were at least ~15,000 bp in size. Only unique CNVs (i.e., observed once only) were considered to eliminate multi-allelic CNVs that existed in the parents and were unlikely to be de novo mutations. Lastly, CNVs were identified as germline mosaic when they were present in a portion of the siblings from the same sire, but not in any offspring from a different sire.

**CNV Validation.** Real-time PCR using TaqMan® Copy Number Assays that targeted the CNV regions were analyzed using a CFX96™ Real-Time System (Bio-Rad, Saint-Laurent, Quebec, Canada) to validate putative de novo CNVs. The mouse transferrin receptor gene (Tfrc) was used as a reference and analyzed simultaneously with the CNV targeting assays. Family quartets (sire, dam, offspring with CNV, offspring without CNV) were analyzed together, and each tissue was analyzed with at least four technical replicates. The same DNA used in the arrays from the liver (endoderm) was used for validation. DNA from the brain (ectoderm) or spleen (mesoderm) was also analyzed for the presence of the CNV in the offspring to confirm that the CNV was germline in origin and not a result of somatic mosaicism. CNVs that adhered to Mendelian inheritance patterns or had a copy number change of <0.5 copies were considered as false positives.

Out of the initial call-set of 30 de novo CNVs (28 unique, two mosaic) that passed the threshold criteria, 12 were eliminated (six false positives, six inherited from parent). In four of the six false positives, the relative PCR amplification of mutated offspring liver DNA compared with parental DNA was concordant with array data; however, the absolute copy number change was less than <0.5 copies. Thus, these CNVs were more likely to be somatic mosaics that arose during early embryogenesis as opposed to those in the germ cells[51]. Of the 18 remaining CNVs, one failed to amplify after testing multiple assays and was likely a somatic mosaic based on the array results, 17 CNVs were considered de novo (15 CNVs) or germline mosaic events (two CNVs). In all cases where SNP information was available, the CNV was determined to have arisen from the paternal genome (seven of 17 CNVs). However, there were two deletions (log$_2$ ratio of ≥ −0.75), where neither parental allele dropped out, indicating that the CNVs were somatic events occurring after fertilization. Therefore, in order for a deletion to be called as a true de novo mutation the log$_2$ ratio needed to be < −0.75.

**Identifying CNV breakpoint and parental origin.** Mate-pair libraries were built for each animal carrying a CNV (Supplementary Data 6) using the Nextera mate pair library prep kit (Illumina®, Vancouver, British Columbia, Canada). Each library was sequenced using a NextSeq 500 (Illumina®) sequencer and ~20% of a high-output flow cell. Following alignment to the mm10 genome using bwa (version 0.7.12), the regions identified by aCGH were visually inspected using the Integrative Genomics Viewer[46] to pinpoint the CNV breakpoint sequence and elucidate the mechanism for formation. The sequence data were also compared against the MutaMouse and C57BL/6J haplotype information to characterize the parental origin of each CNV. Loss of the allele in deletions or higher proportion of the allele in duplications would indicate that the CNV originated in a germ cell of the parent carrying that allele.

**Statistical analyses.** Assuming a Poisson distribution for the error, GEEs were used to analyze SNV/indel data. One-tailed Fisher's exact tests were used to compare CNV events between the control and BaP animals. The Poisson distribution was used for SNV data, and the modified Wald-method was used to measure the 95% confidence interval for CNV data. The Holm–Sidak adjustment was used to control for multiple comparisons. Cosmic Signature analysis was undertaken using SomaticSignatures[52] and deconstructSigs packages[53] in R[54].

**Reporting summary.** Further information on research design is available in the Nature Research Reporting Summary linked to this article.

## Data availability
The raw whole-genome sequences and the array data generated for this study are available in the Sequence Read Archive under BioProject No. PRJNA508974. All other data are available from the corresponding authors on reasonable request.

## Code availability
Custom scripts can be accessed in the GitHub repository: https://github.com/mattjmeier/HC/tree/master/BaP_pedigree. These were used to identify Mendelian violations between offspring and parents from genomic variant call files generated by the Torrent Variant Caller.

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

## Acknowledgements

This work was funded by the Genomics Research and Development Initiative. We would like to thank the Mechanistic Studies Division at Health Canada for their support of this project and the Scientific Services Division for help with animal exposure, maintenance, and necropsies.

## Author contributions

M.A.B. conducted the animal study, collected samples, performed whole genome sequencing and array CGH, and contributed to data analyses. M.M. and R.G. contributed to sequencing and whole-genome analyses. A.R.C. conducted array CGH validation of copy number variants. S.J.L. and T.F. performed validation of candidate de novo mutations and advised on bioinformatics pipeline. A.W. contributed to statistical analyses of whole genome and array CGH data. M.E.H. consulted and advised on study design and data analyses. C.Y. and F.M. secured the funding for the study and were

responsible for study conception, study design, and results interpretation. All authors contributed to data interpretation, paper writing, and approved the final version.

## Additional information

**Competing interests:** The authors declare no competing interests.

