## [Peer Review File · Communications Biology]

Reviewers' comments:

Reviewer #1 (Remarks to the Author):

In the manuscript „Paternal exposure to benzo(a)pyrene induces genome wide mutations in mouse offspring” by Beal and colleagues the authors used next gen sequencing and comparative genomic hybridization assays to investigate genome-wide mutations in the offspring of male mice exposed to the environmental mutagen BaP. In principle the study is interesting, however in its present form it is not suitable for publication. From what is judgeable by this reviewer, who is not an expert in DNA sequencing, the study is well-designed, and the data appear to be of suitable quality. However, the text of the manuscript is very hard to follow for a non-expert in genome sequencing, and the informational content of the figures is rather low. Before publication of the manuscript, the authors need to put considerable effort in improving the reader-friendliness of the manuscript and the quality of their data presentation. An example of a similar study, in which this worked out nicely, is the publication by Adewoye et al. (Nature Comm 2015). To start with, the authors may consider including a scheme explaining their experimental design. Furthermore, e.g., it must be clearly stated in the main text, from which organ DNA was used for the analyses and what was the rationale behind the choice. Moreover, it needs to be specified what is meant by ‘sample size’ in Table 1. Also, it is unclear what ‘BaP + 3’ or ‘BaP +42’ means in Figure 2. Just to give a few examples, how text and display items could be improved.

From a toxicological point of view, the study would be strengthened if dose-response data were included, i.e. treating mice with different doses of BaP. Does this lead to a correlation of BaP doses and mutation frequencies? Does a threshold dose of BaP exist below which no mutations are induced in offspring? Furthermore, the study would benefit from additional (quantitative) bioanalytical data (e.g., by isotope dilution mass spectrometry) demonstrating BPDE-DNA adducts in male germ cells after BaP exposure (i.e., as biomarkers of exposure). These data could be also correlated to mutation frequencies observed in the offspring and could be useful to address the question of the existence of a threshold dose. Finally, it would be very interesting to characterize the offspring in much greater detail, with regards to, e.g., cancer development, disease and aging phenotypes.

Reviewer #2 (Remarks to the Author):

The authors investigated how the exposure of male sperm to benzo(a)pyrene affects offspring in terms of de novo point mutations and de novo copy number changes. The authors perform arrayCGH and WGS to identify these de novo events and confirm identified events by RT-PCR and targeted resequencing. The authors conclude that offspring show increased de novo duplications, but not deletions, compared to controls, as well as an increased number of de novo SNVs. The authors also find that post-mitotic exposure results more in embryonic mutations but not de novo mutations, in line with their hypothesis.

The is a well-designed study with interesting findings that may shed light on what kind and how new mutations arise due to exposure to DNA damaging agents and the effect on the DNA of offspring. The manuscript is at times difficult to follow, because of the various different conditions and types of variation. In addition, I have concerns about some of the results in the manuscript.

Major comments

1. It would be useful for the readers if the authors included a diagram of the experimental design

indicating the different experiments, treatment groups, time points and for which mice arrayCGH and WGS was performed. In addition the authors could try to indicate per section more clearly what condition, and variant types are being discussed.

2. De novo duplications

2.1 The numbers on which the increased de novo duplication events were based on are very low and hence the statistical p-value is only just significant (actually $p=0.03$ according to my calculation. R: `fisher.test(matrix(c(6, 181-6, 0, 171),2,2))`).

2.2 The numbers initially mentioned in the text (6 events in 181 mice) do not match those of table 1 (5 events in 181 mice).

2.3 The authors should only mention the corrected p-values rather than uncorrected values. In this case especially it turns out that only the uncorrected value is significant according to the 0.05 threshold.

2.4 Although the authors take into account whether events are identified in the same mice, they do not make clear whether mice with de novo duplications are siblings or not. This should be taken into account. The calculation based on sires is a better approach but although a p-value is mentioned it is unclear how this was calculated, and whether this value has been corrected for multiple testing.

2.5 In any case, the result, as it is, is not fully convincing and does not seem to support strong conclusions without greater numbers or replication. The authors should indicate this clearly and I would suggest making this result of their study less prominent.

2.6 The authors hypothesize that microhomology may be a model that explains the occurrence of the duplication events. It seems that these microhomologies are often as small as 2 bp, leaving the reader in doubt whether the microhomologies are merely coincidence or not. The authors should quantify the presence of microhomologies statistically, or otherwise clearly indicate that they are speculating and there is no significant enrichment for microhomology at the CNV breakpoints.

3. Germline and somatic mosaic CNVs

3.1 The authors report the identification of 2 de novo deletions (mosaic in parents) of which at least one is likely maternal in origin. As such it seems that no conclusions in relation to the BaP exposure can be drawn from this and I don't understand the relevance to the current study.

3.2 For the somatic mosaic deletions the authors find many events. It remains unclear how many of these were validated independently?

3.3 Are these the events mentioned also in Table 1. It only says Embryonic SNVs/indels, not Embryonic CNVs? Are these events found in controls as well? How are these events relevant to the current study?

3.4 The authors state that large deletions occur at high rate during development. Compared to what? To humans?

3.5 Overall, it is unclear to me how the results in this section (although interesting) are connected the main message / hypotheses of the manuscript. If they are not, I would suggest the authors add these results at the end of the manuscript in a discussion style or remove them from the manuscript.

4. De novo SNVs and indels

4.1 The authors should mention the test used for comparing de novo SNVs between treated and control animals. Does the test take into account litter specific effects?

4.2 Please indicate clearly on which mutations mutational signature analysis was performed. I assume the germline de novo SNVs? Or also on the embryonic mutations?

4.3 This seems to be the strongest results of the authors. The finding that only some offspring are affected is very interesting and worth further exploration (in other studies). The authors should consider treating this result first in the manuscript.

Minor comments

The authors could consider a somewhat broader introduction on BaP and how it interacts with DNA, and their choice for this mutagen in particular.

Line 42, the authors now only mention paternal age as a mutagen, but could consider mentioning others such as maternal age, (e.g. Wong et al. Nat comm. 2016), ionizing radiation (e.g. Adewoye et al. Nat Comm., 2015), and dioxin (Ton et al. Hum. Mutat. 2018)

Line 145, the word "and" should be removed

Line 155. Please make more clear that MMBI and replication fork collapse are related

Line 477, two-tailed tests should be used as a default for testing.

Line 477, Please clarify "GEEs"

Table S1, please indicate more clearly using separate columns, the experimental group in which the event was identified, and whether the event was germline or embryonic. The column "possible disease outcome" is I think not very relevant here, and is somewhat confusing, as the mice were not tested for diseases.

Line 245, "these extra mutations". I assume that the analysis was not performed on "extra" mutations, but on all mutations of the animals with the highest number of mutations.

The authors should include all identified de novo SNVs as a supplementary to the manuscript, allowing readers to reproduce their analysis of mutational signatures.

Reviewer #3 (Remarks to the Author):

In the manuscript, Beal MA and colleagues report the detection of benzo(a)pyrene (BaP)-induced germline mutations in mouse using CGH-array and whole genome sequencing (WGS) methods. By controlling the timing of mating after BaP administration, they observed the mutations caused by either mitotic stage specific BaP exposure or post-mitotic stage specific BaP exposure, independently. According to the results, the authors discuss the effect and mechanism of intergenerational carryover effect of BaP-adduct. Although the post-mitotic exposure of BaP is likely to cause embryonic mutation with mosaicism via DNA adduct in sperm, supporting evidence and logic seem weak yet. Similar to chemical mutagen (such as ENU) induced- and radiation induced-germline mutation experiments, BaP-induced germline mutation experiment will be an example for the environmental cause of germline mutation in mammals.

Comments.

To discuss the effect of BaP, author should mention that

- (1) How toxic is the dose of BaP administered in this experiment on cell as well as on mouse.
- (2) If the BaP adduct blocks DNA replication, 1000 of BaP adducts might be lethal for the cell. Do authors think that all of the BaP adducts are removed in a fertilized egg?
- (3) Does the fertilized egg (and also spermatogonia) possess DNA repair enzymes for TLS, NER and

MMBIR?

As mentioned by authors (lines 241-243), there was a variation among the BaP-treated animals. Were the BaP treatments performed in properly? If the variation is caused by technical reason (the degree of BaP exposure on the testis was not equal among the mice), the mutation rate and other calculations related to the BaP treatment are not appropriate for conclusion.

Minor comments

Fisher's exact test

lines 113-114, Is $P = 0.04$ correct? (calculation)

lines 112-114, Authors should discuss the comparison between control (0/171) vs mitotic exposure (4/83) and control (0/171) vs post-mitotic exposure (2/98) , too.

line 247, What the P-value indicates the difference between?

COMMSBIO-19-0050

Beal et al - Paternal exposure to benzo(a)pyrene induces genome-wide mutations in mouse offspring

Responses to reviewers

We thank the reviewers for the very constructive and helpful comments that they have provided. We believe that the revised manuscript is much improved because of their input.

Below we provide our responses to the reviewers' comments. The line numbers in our responses refer to the line numbers in the version of the manuscript with track changes. Also, yellow highlight in the version of the manuscript with track changes identifies text that was already present in the original submission and that has been moved to a different section due to the manuscript reorganization that we have made in response to the reviewers' comments.

Reviewer #1 (Remarks to the Author):

In the manuscript "Paternal exposure to benzo(a)pyrene induces genome wide mutations in mouse offspring" by Beal and colleagues the authors used next gen sequencing and comparative genomic hybridization assays to investigate genome-wide mutations in the offspring of male mice exposed to the environmental mutagen BaP. In principle the study is interesting, however in its present form it is not suitable for publication. From what is judgeable by this reviewer, who is not an expert in DNA sequencing, the study is well-designed, and the data appear to be of suitable quality.

We thank the reviewer for the positive assessment of our study.

However, the text of the manuscript is very hard to follow for a non-expert in genome sequencing, and the informational content of the figures is rather low. Before publication of the manuscript, the authors need to put considerable effort in improving the reader-friendliness of the manuscript and the quality of their data presentation. An example of a similar study, in which this worked out nicely, is the publication by Adewoye et al. (Nature Comm 2015).

We have simplified the manuscript by improving the figures, reorganizing and simplifying the text, and moving materials and findings that are not essential to the objectives of the study to supplementary information. We believe that these changes have improved the readability of the manuscript for non-specialists.

To start with, the authors may consider including a scheme explaining their experimental design.

We agree. A new Figure 1 showing the experimental design has been added to the manuscript.

Furthermore, e.g., it must be clearly stated in the main text, from which organ DNA was used for the analyses and what was the rationale behind the choice.

The organ DNA used is now mentioned in the main text for both NGS and aCGH analyses. Specifically, in the second paragraph of the Results and Discussion section (lines 118 and 138 of revised manuscript) and, then again in the Materials and Methods (lines 677 and 738 of revised manuscript).

Moreover, it needs to be specified what is meant by 'sample size' in Table 1.

A footnote has been added to Table 1 to clarify what we mean by sample size. Specifically: " Sample size refers to the number of offspring analyzed in each respective group."

Also, it is unclear what 'BaP + 3' or 'BaP +42' means in Figure 2. Just to give a few examples, how text and display items could be improved.

We have replaced BaP +3 and BaP +42 with post-mitotic and mitotic in the figure itself and have added the following text to the figure legend: "Post-mitotic and mitotic refer to matings occurring 3 and 42 days after the end of exposure, respectively."

From a toxicological point of view, the study would be strengthened if dose-response data were included, i.e. treating mice with different doses of BaP. Does this lead to a correlation of BaP doses and mutation frequencies? Does a threshold dose of BaP exist below which no mutations are induced in offspring?

The reviewer has certainly identified some important questions. The present study had two main objectives: (1) determine whether paternal exposure to BaP induces mutations in the offspring; and (2) determine whether the magnitude of the response differs depending on the exposed phase of spermatogenesis. The results show that paternal exposure to BaP does induce mutations in the offspring and that both exposure windows of spermatogenesis are equally susceptible to the induction of mutations. We believe that the present manuscript stands on its own without the need of additional dose-response studies. However, we agree with the reviewer that an important next step is establishing the dose-response relationship between exposure and mutations and we have added this to the conclusions of the manuscript.

Furthermore, the study would benefit from additional (quantitative) bioanalytical data (e.g., by isotope dilution mass spectrometry) demonstrating BPDE-DNA adducts in male germ cells after BaP exposure (i.e., as biomarkers of exposure). These data could be also correlated to mutation frequencies observed in the offspring and could be useful to address the question of the existence of a threshold dose.

The presence of DNA adducts in male germ cells after exposure to BaP has been previously reported (Verhofstad et al., 2010, Environ Mol Mutagen 51:123-129). As indicated in lines 73-75 of the revised manuscript, we have data on sperm mutant frequencies (O'Brien et al., 2016, Toxicol Sci 152:363-371) and microsatellite mutations (Rowan-Carroll et al., 2017, Mutagenesis 32:463-470) from mice that had been exposed to BaP at the same time as those used for breeding in the present study. We demonstrate through these studies that there was sufficient exposure to cause mutations in *lacZ* and microsatellites in the sperm of these mice, consistent with the expected genetic effects of the exposure. Thus, we do not feel that additional measures of exposure are needed at this time.

Finally, it would be very interesting to characterize the offspring in much greater detail, with regards to, e.g., cancer development, disease and aging phenotypes.

We agree that this would be very interesting. However, we point out that such a study would require a large number of animals that would need to be followed for the duration of their lifetime. As such, we think that this is beyond the purpose of the present study.

Reviewer #2 (Remarks to the Author):

The authors investigated how the exposure of male sperm to benzo(a)pyrene affects offspring in terms of de novo point mutations and de novo copy number changes. The authors perform arrayCGH and WGS to identify these de novo events and confirm identified events by RT-PCR and targeted resequencing. The authors conclude that offspring show increased de novo duplications, but not deletions, compared to controls, as well as an increased number of de novo SNVs. The authors also find that post-mitotic exposure results more in embryonic mutations but not de novo mutations, in line with their hypothesis.

This is a well-designed study with interesting findings that may shed light on what kind and how new mutations arise due to exposure to DNA damaging agents and the effect on the DNA of offspring. The manuscript is at times difficult to follow, because of the various different conditions and types of variation. In addition, I have concerns about some of the results in the manuscript.

We thank the reviewer for the positive assessment. As already indicated in the response to reviewer 1, we have tried to improve the readability of the manuscript by improving the figures, reorganizing and simplifying the text, and moving to supplementary materials findings that are not essential to support the major findings of the study.

Major comments

1. It would be useful for the readers if the authors included a diagram of the experimental design indicating the different experiments, treatment groups, time points and for which mice arrayCGH and WGS was performed. In addition the authors could try to indicate per section more clearly what condition, and variant types are being discussed.

Reviewer 1 had also commented on the need for a Figure showing the experimental design. We agree and we have added a new Figure 1 in the manuscript. Also, we have gone through the manuscript and clarified as much as possible what type of variant is being discussed.

2. De novo duplications

2.1 The numbers on which the increased de novo duplication events were based on are very low and hence the statistical p-value is only just significant (actually $p=0.03$ according to my calculation. R: `fisher.test(matrix(c(6, 181-6, 0, 171),2,2))`).

CNV mutations are indeed rare events. However, at the population level and given the number of base pairs affected, these are very important biological events. Also, we note that we have used approximately 350 mice in this analysis and that this is the largest of such studies to have been published to date on this topic. For example, previous work by Adewoye et al. on radiation analyzed approximately 250 mice.

Our Fisher's Exact Test value was slightly different from that obtained by the reviewer because we used 5 CNVs instead of 6 as the reviewer has done. We have clarified that in lines 313-314 of the revised manuscript: "Two of the duplication CNVs occurred in the same animal, thus the total number of CNVs was conservatively counted as 5". Furthermore, this analysis is impacted by comment 2.3 below.

2.2 The numbers initially mentioned in the text (6 events in 181 mice) do not match those of table 1 (5 events in 181 mice).

As noted above, we have added a sentence to the main text and a footnote in the table to clarify the difference between 5 and 6 CNVs events.

2.3 The authors should only mention the corrected p-values rather than uncorrected values. In this case especially it turns out that only the uncorrected value is significant according to the 0.05 threshold.

As the reviewer is suggesting, only adjusted p values are now reported in the text. In addition, following the reviewer's comment, we have consulted with the statistician on the project and instead of reporting the data as simply using the number of males that had been exposed to BaP, as it was done in the original manuscript, we are not reporting the frequencies of duplication CNVs taking into account the number of offspring analyzed for each sire. Specific to this point, the text (lines 313-343 of the revised manuscript) was modified to read: "Two of the duplication CNVs occurred in the same animal, thus, the total number of CNVs was conservatively counted as five. Using this number, the increase in duplications in the BaP group was at the borderline of significance (Fisher's exact, adjusted $P=0.08$). However, the five BaP-induced duplication events occurred in five different litters from four sires (males 32, 33, 37 and 39). Male 32 had an offspring with a duplication CNV at both time points, one involving chromosome 3 and the other chromosome 16; while male 33 had a male offspring with a duplication involving chromosome 5 and a female with a deletion involving chromosome 3 within the same litter (Table S6). Using sires as the biological unit, and considering the number of offspring analyzed for each sire (Table S7), yields a statistically significant increase in the frequency of duplication CNVs (0.0278 vs 0; Fisher Exact Test, adjusted $P<0.001$) demonstrating that this result is not due to clustering within a litter or within a BaP-exposed male."

2.4 Although the authors take into account whether events are identified in the same mice, they do not make clear whether mice with de novo duplications are siblings or not. This should be taken into account. The calculation based on sires is a better approach but although a p-value is mentioned it is unclear how this was calculated, and whether this value has been corrected for multiple testing.

Please see our response above. Also, we have added text in the materials and methods to explain the coding used in the tables to identify the offspring with SNV and CNV mutations (lines 662-666 of the revised manuscript) and the number of males that generated the litters used in this study (lines 657-660 of the revised manuscript).

2.5 In any case, the result, as it is, is not fully convincing and does not seem to support strong conclusions without greater numbers or replication. The authors should indicate this clearly and I would suggest making this result of their study less prominent.

We have noted the concern of the reviewer and have revised the text to introduce a word of caution. However, we think that our finding is quite convincing, given the fact that duplication events have been reported only once among 307 controls, including those from our study and those of Adewoye et al and Arlt et al. We have included the revised text below (lines 343-347) to address the reviewer's comment: "Indeed, the spontaneous frequency of duplications appears to be rare compared to deletions, as only a single duplication was detected from the 93 control animals from Adewoye et al.¹¹, 43 control animals from Arlt et al.³⁴, and 171 animals in this study. Thus, although we have recovered few duplications overall, this increase in the exposed mice is unlikely to be due to chance alone."

2.6 The authors hypothesize that microhomology may be a model that explains the occurrence of the duplication events. It seems that these microhomologies are often as small as 2 bp, leaving the reader in doubt whether the microhomologies are merely coincidence or not. The authors should quantify the presence of microhomologies statistically, or otherwise clearly indicate that they are speculating and there is no significant enrichment for microhomology at the CNV breakpoints.

Extensive evidence in humans has been published to show that the microhomologies are typically extremely small at CNVs breakpoints. Similar levels of homology have been reported in the CNVs of mouse germline following exposure to hydroxyurea (Arlt et al, 2018, Environ Mol Mutagen 59:698-714). Thus, we note that that the occurrence of microhomology at the breakpoint is not a novel finding and we just report that we observe small regions of homology at the breakpoint of BaP-induced CNVs as it has been observed in both human and mouse CNVs. We have cited the appropriate references and we feel that this is thoroughly explained in the text. We direct the reader to the relevant references in the human literature that describe these types of events in detail (lines 398-406 of revised manuscript). Also, we have not conducted statistical testing because as shown in Table S6, all de novo CNVs have microhomology at the breakpoint. The majority of germline mosaics and embryonic CNVs do as well (Table S6). It is clearly the predominant mechanism.

3. Germline and somatic mosaic CNVs

The reviewer has raised several questions about germline and somatic mosaicism CNVs. Comments 3.1 to 3.4 are secondary to the main comment expressed in 3.5, thus, we address this comment first.

3.5 Overall, it is unclear to me how the results in this section (although interesting) are connected to the main message / hypotheses of the manuscript. If they are not, I would suggest the authors add these results at the end of the manuscript in a discussion style or remove them from the manuscript.

We agree with the reviewer that these interesting results are not related to the exposure and are not necessarily supporting the main conclusions of the manuscript. Thus, we have shifted the bulk of the results originally presented in this section to supplementary materials. We have retained a sentence (lines 429-430) indicating that these additional findings are described in supplementary materials.

3.1 The authors report the identification of 2 de novo deletions (mosaic in parents) of which at least one is likely maternal in origin. As such it seems that no conclusions in relation to the BaP exposure can be drawn from this and I don't understand the relevance to the current study.

We agree and have removed the discussion of these events from the main manuscript.

3.2 For the somatic mosaic deletions the authors find many events. It remains unclear how many of these were validated independently?

We did not validate these events.

3.3 Are these the events mentioned also in Table 1. It only says Embryonic SNVs/indels, not Embryonic CNVs? Are these events found in controls as well? How are these events relevant to the current study?

These events are not included in Table 1; however, they are included in Supplementary table 6 (old supplementary table 1).

3.4 The authors state that large deletions occur at high rate during development. Compared to what? To humans?

The statement has been deleted.

4. De novo SNVs and indels

4.1 The authors should mention the test used for comparing de novo SNVs between treated and control animals. Does the test take into account litter specific effects?

The test used (Generalized estimating equations) has been added to the main text (lines 182-183) and we have clarified that it takes into account individual effects rather than litter specific effects.

4.2 Please indicate clearly on which mutations mutational signature analysis was performed. I assume the germline de novo SNVs? Or also on the embryonic mutations?

Mutational signature analysis was performed on total SNVs (germline and embryonic). The text has been updated to clarify this (lines 215-216 of the revised manuscript).

4.3 This seems to be the strongest results of the authors. The finding that only some offspring are affected is very interesting and worth further exploration (in other studies). The authors should consider treating this result first in the manuscript.

We agree with the reviewer that these SNV findings should be presented first and have re-ordered the manuscript based on this suggestion.

Minor comments

The authors could consider a somewhat broader introduction on BaP and how it interacts with DNA, and their choice for this mutagen in particular.

We agree and have added some text to describe BaP and how it interacts with DNA (lines 69-73 of revised manuscript): "BaP is an IARC Class 1 carcinogen that is metabolically activated by cytochrome p450s to produce reactive metabolites, such as 7,8-dihydrodiol-9,10-epoxide (BPDE)³⁴, that form bulky DNA adducts. These DNA lesions can cause strand breaks, and replication of DNA containing unrepaired adducts leads primarily to G to T mutations¹³."

Line 42, the authors now only mention paternal age as a mutagen, but could consider mentioning others such as maternal age, (e.g. Wong et al. Nat comm. 2016), ionizing radiation (e.g. Adewoye et al. Nat Comm., 2015), and dioxin (Ton et al. Hum. Mutat. 2018)

We agree; however, we note that the Adewoye et al paper is in mice, thus, we have not included it in this section that relates to human germ cell mutagens. We have added the following sentence to incorporate the other references mentioned by the reviewer (lines 47-50 of the revised manuscript): "Similarly, more recent evidence suggests that maternal age⁷ and dioxin⁸ also increase the mutational burden in humans. These findings emphasize the need to identify both the endogenous and exogenous factors that may be contributing to human germ cell mutagenesis and genetic diseases."

Line 145, the word "and" should be removed

Done.

Line 155. Please make more clear that MMBI and replication fork collapse are related

The text was revised as suggested (see line 413 of revised manuscript).

Line 477, two-tailed tests should be used as a default for testing.

We respectfully disagree with the reviewer. There is no biological reason why we would expect this exposure to cause a decrease in CNVs. The hypothesis that we are testing is that BaP increases CNVs mutation rates. Thus, a one-tail test is appropriate. Also, we now discuss only adjusted p-values.

Line 477, Please clarify "GEEs"

The text was revised as suggested.

Table S1, please indicate more clearly using separate columns, the experimental group in which the event was identified, and whether the event was germline or embryonic. The column "possible disease outcome" is I think not very relevant here, and is somewhat confusing, as the mice were not tested for diseases.

We agree and have revised the table, which is now Table S6, as suggested.

Line 245, "these extra mutations". I assume that the analysis was not performed on "extra" mutations, but on all mutations of the animals with the highest number of mutations.

The reviewer is correct. We have modified the text (lines 206-212 of the revised manuscript) to say: "The mutation spectra of the seven offspring with the highest numbers of mutations (Figure 2) revealed that these extra mutations were consistent with the mutation spectrum of BaP (i.e., increased frequency of G:C  T:A, G:C  C:G) observed in the sperm and bone marrow of exposed males^{14,27}. Furthermore, there was also an increase in the number of G:C  A:T mutations (Figure 2). Overall, the mutation spectrum in the affected animals in the exposed group was significantly different than that observed in unaffected animals and controls (Fisher's Exact P = 0.0009)."

The authors should include all identified de novo SNVs as a supplementary to the manuscript, allowing readers to reproduce their analysis of mutational signatures.

We have done as suggested. All identified SNVs are now included as supplementary Table S5.

Reviewer #3 (Remarks to the Author):

In the manuscript, Beal MA and colleagues report the detection of benzo(a)pyrene (BaP)-induced germline mutations in mouse using CGH-array and whole genome sequencing (WGS) methods. By controlling the timing of mating after BaP administration, they observed the mutations caused by either mitotic stage specific BaP exposure or post-mitotic stage specific BaP exposure, independently. According to the results, the authors discuss the effect and mechanism of intergenerational carryover effect of BaP-adduct. Although the post-mitotic exposure of BaP is likely to cause embryonic mutation with mosaicism via DNA adduct in sperm, supporting evidence and logic seem weak yet. Similar to chemical mutagen (such as ENU) induced- and radiation induced-germline mutation experiments, BaP-induced germline mutation experiment will be an example for the environmental cause of germline mutation in mammals.

Comments.

To discuss the effect of BaP, author should mention that

(1) How toxic is the dose of BaP administered in this experiment on cell as well as on mouse.

The dose used in this study was 100 mg/kg/day, which was selected based on the results of a pilot study that indicated that a higher dose would have resulted in significant toxicity and morbidity to the animals. This is now noted in the materials and methods (lines 643-645 of the revised manuscript).

(2) If the BaP adduct blocks DNA replication, 1000 of BaP adducts might be lethal for the cell. Do authors think that all of the BaP adducts are removed in a fertilized egg?

As noted in the discussion, the exposure is likely to result in as many as 1000 adducts per genome in the exposed sperm, however, there was no evidence that this level of adducts caused a reduction in fertilization and/or embryonic development because the litter sizes were similar in exposed and control groups. We have now added this information to the beginning of the results/discussion section.

Also, as described in the second paragraph of the results/discussion, because of a decline in DNA repair capabilities during the post-mitotic phase spermatogenesis, we expect to see adducts retained in the fertilizing sperm only at the early mating time. It is established that sperm fertilizing eggs at this exposure time point would have adducts that would need to be repaired by the egg. Although we know that oocytes are DNA repair competent it is highly unlikely that the eggs would be able to remove all of these adducts before the initial rounds of embryonic DNA replication begin. This is described in lines 103-108 of the revised manuscript and, indeed, it is what we observed.

(3) Does the fertilized egg (and also spermatogonia) possess DNA repair enzymes for TLS, NER and MMBIR?

As indicated in our response above, we have added information about the DNA repair capacity of the oocyte. However, as far as we know, there no available data in mammalian oocytes or spermatogonia regarding MMBIR.

4) As mentioned by authors (lines 241-243), there was a variation among the BaP-treated animals. Were the BaP treatments performed in properly? If the variation is caused by technical reason (the degree of BaP exposure on the testis was not equal among the mice), the mutation rate and other calculations related to the BaP treatment are not appropriate for conclusion.

The exposure was done by expert staff in the Health Canada's animal resources division. We are confident that the exposures were conducted properly. Also, we note that the observed variability in mutations in the offspring occurred within and not across litters. Thus, we are confident that this is true internal biological variability. This has been further clarified in the text (lines 196-188 of the revised). "Moreover, there was a large degree of variation among the offspring of BaP-treated males, with seven of the mice having numbers of mutations outside the range of control mutations and five mice being within the control range. This observed variability in mutations occurred within litters, indicating that this was not due to differences in exposure levels among sires."

Minor comments

Fisher's exact test. lines 113-114, Is $P = 0.04$ correct? (calculation)

As noted in our response to reviewer #2, $P = 0.04$ is correct because two of the CNVs occurred in the same animal and were conservatively counted as one event. However, we are now reporting the adjusted p value as requested by reviewer #2.

lines 112-114, Authors should discuss the comparison between control (0/171) vs mitotic exposure (4/83) and control (0/171) vs post-mitotic exposure (2/98) , too.

We have not done so because we do not have the power to separate the analysis into these two subgroups.

line 247, What the P-value indicates the difference between?

We have revised the text (lines 210-212 of the revised manuscript) to state: " Overall, the mutation spectrum in the affected animals in the exposed group was significantly different than that observed in unaffected animals and controls (Fisher's Exact $P = 0.0009$)."

REVIEWERS' COMMENTS:

Reviewer #1 (Remarks to the Author):

The revised manuscript by Beal et al. has significantly improved in terms of readability and data presentation. All my editorial suggestions have been addressed and incorporated in the revised manuscript. My suggestions for additional experiments have not been addressed experimentally, however have been considered by the authors in the 'discussion' and 'conclusions' parts of the manuscript. I recommend publication of the manuscript.

Reviewer #2 (Remarks to the Author):

The authors have convincingly addressed all my concerns in their rebuttal.

I congratulate the authors on a very interesting manuscript!